# Genome-Wide Identification and Expression Analysis of MAPK Gene Family in Lettuce (*Lactuca sativa* L.) and Functional Analysis of *LsMAPK4* in High- Temperature-Induced Bolting

**DOI:** 10.3390/ijms231911129

**Published:** 2022-09-22

**Authors:** Tingzhen Wang, Mingjia Liu, Yang Wu, Yufeng Tian, Yingyan Han, Chaojie Liu, Jinghong Hao, Shuangxi Fan

**Affiliations:** Beijing Key Laboratory of New Technology in Agricultural Application, National Demonstration Center for Experimental Plant Production Education, Plant Science and Technology College, Beijing University of Agriculture, Beijing 102206, China; wtz1398932310@163.com (T.W.); a1997lmj@163.com (M.L.); 18612006362@163.com (Y.W.); t18419067134@126.com (Y.T.); hyybac@126.com (Y.H.); cliu@bua.edu.cn (C.L.)

**Keywords:** MAPK, lettuce, genome-wide analysis, bioinformatics analysis, high temperature, expression pattern, bolting, virus-induced gene silencing, *LsMAPK4*

## Abstract

The mitogen-activated protein kinase (MAPK) pathway is a widely distributed signaling cascade in eukaryotes and is involved in regulating plant growth, development, and stress responses. High temperature, a frequently occurring environmental stressor, causes premature bolting in lettuce with quality decline and yield loss. However, whether MAPKs play roles in thermally induced bolting remains poorly understood. In this study, 17 *LsMAPK* family members were identified from the lettuce genome. The physical and chemical properties, subcellular localization, chromosome localization, phylogeny, gene structure, family evolution, cis-acting elements, and phosphorylation sites of the *LsMAPK* gene family were evaluated via in silico analysis. According to phylogenetic relationships, *LsMAPKs* can be divided into four groups, A, B, C, and D, which is supported by analyses of gene structure and conserved domains. The collinearity analysis showed that there were 5 collinearity pairs among *LsMAPKs*, 8 with *AtMAPKs*, and 13 with *SlMAPKs*. The predicted cis-acting elements and potential phosphorylation sites were closely associated with hormones, stress resistance, growth, and development. Expression analysis showed that most *LsMAPKs* respond to high temperatures, among which *LsMAPK4* is significantly and continuously upregulated upon heat treatments. Under heat stress, the stem length of the *LsMAPK4*-knockdown lines was significantly shorter than that of the control plants, and the microscope observations demonstrated that the differentiation time of flower buds at the stem apex was delayed accordingly. Therefore, silencing of *LsMAPK4* significantly inhibited the high- temperature-accelerated bolting in lettuce, indicating that LsMPAK4 might be a potential regulator of lettuce bolting. This study provides a theoretical basis for a better understanding of the molecular mechanisms underlying the MAPK genes in high-temperature-induced bolting.

## 1. Introduction

Mitogen-activated protein kinase (MAPK) is a serine/threonine protein kinase [1]. Through stepwise phosphorylation, MAPK, MAPKK (mitogen-activated protein kinase kinase), and MAPKKK (mitogen-activated protein kinase kinase kinase) collectively form a highly conserved MAPKKK → MAPKK → MAPK cascade, signal-transduction pathway [2]. When MAPKKK is activated by a sensor/receptor it can phosphorylate serine/threonine residues of S/TXXXXXS/T, in which X represents any amino acid, to activate downstream MAPKK, after which MAPKK continues to phosphorylate the conserved threonine and tyrosine residues in the TXY motif downward to activate MAPK [3,4,5,6,7,8]. MAPK cascades respond to almost all biological or abiotic stresses [9,10,11], such as mechanical damage [12], high salt [13], low temperature [14], drought [15] and pathogen invasion [16] and widely participate in the process of plant growth and development [17,18].

MAPK is located downstream of the entire MAPK signaling cascade. After phosphorylation, MAPK has three targets: cytoplasmic protein kinases or transcription factors (TFs), cytoplasmic cytoskeleton, and nuclear transcription factors [19]. Many plant MAPK genes have been studied, and sequence analysis of the whole genome showed that *Arabidopsis thaliana* has 20 MAPK genes, which can be separated into four groups, A, B, C, and D, according to their phosphorylation motif. The phosphorylation motif of groups A, B, and C was TEY, and that of group D was TDY [9,20]. The C-terminus of MAPK in groups A, B, and C contains the CD domain, which is speculated to be the binding region of upstream MAPKK, phospholipase, and various downstream substrates. The CD domain of group C is modified. Group D does not contain this region [21,22]. In addition to *Arabidopsis*, 15 MAPK genes were identified in rice [23], 11 in chrysanthemum [24], 43 in cultivated strawberry [25], and 12 in grape [26]. The functions of many MAPK genes have been identified and are known to participate in almost all activities in the plant life cycle, including plant growth and development, response to stress signals, activation of stress resistance gene expression, and improvement in adaptability to adversity [5]. MAPKs play an important role in biotic and abiotic stress responses. *Arabidopsis AtMAPK4* has been reported to negatively regulate plant disease resistance and hypertonic resistance [27,28]. In addition, MAPKs are widely involved in hormone synthesis and signal transduction. *AtMAPK6* is involved in regulating the polar transport of auxin and promotes ethylene synthesis together with *AtMAPK3* [29,30]. Ethylene/jasmonic acid (JA) and MAPK3/MAPK6 cooperate to regulate plant defense [31,32]. ABA induces the expression of the MKK3–MAPK6–MYC2 cascade and actively regulates ABA biosynthesis and signal gene transcription [33]. Soybean *GmMAPK4s* negatively regulates salicylic acid accumulation and defense [34]. More importantly, MAPKs are largely involved in plant growth and development. Rice *OsMAPK2* is involved in flower development, and *AtMAPK4* is required for cell plate formation and cell division of male-specific meiosis in *Arabidopsis* [35]. The model *TES/STUD/AtNACK2-AtANPs-AtMKK6-AtMPK4* was proposed to regulate cell meiosis [36,37]. In addition, MAPK also plays an important role in anther development [38,39], ovule development [40], pollen development [17], and seed development [41].

Lettuce is rich in nutrients and is an important component in modern diet and nutrition [42]. Lettuce is sensitive to high temperatures, which can induce premature bolting, resulting in poor growth and reductions in yield and quality. The symbol of bolting is flower bud differentiation, which is the process of rapid stem elongation after flower bud differentiation at the stem tip, accompanied by the transformation from vegetative growth to reproductive growth [43,44]. Bolting is restricted by its development and environment, such as hormones, temperature, and light, and its genetic characteristics, such as cell division, elongation, differentiation, and aging [45,46]. Moreover, the specific regulatory mechanism of high-temperature bolting in lettuce remains unclear, and MAPK has not been reported to play a role in the high temperature-induced bolting of lettuce. Early in this research, we screened differentially expressed MAPK cascade members involved in lettuce bolting at high temperatures, including LsMAPK3, LsMAPK4, LsMAPK9, and LsMAPKK5 protein, using comparative proteomics [47]. Recognizing the crucial role of MAPK in plant growth, development, and stress response, we conducted a genome-wide screening, identification, and bioinformatics analysis of the MAPK gene family in lettuce, determined the differential expression of the MAPK gene family at high temperature, and conducted a virus-induced gene silencing (VIGS) evaluation on *LsMAPK4* with significant differential expression.

## 2. Results

### 2.1. Identification, Chromosomal Localization, and Physicochemical Property Analysis of LsMAPK Family Members in Lettuce

By homology comparison and domain analysis, we identified a total of 17 dependable *LsMAPK* gene family members in the lettuce genome. According to the unified nomenclature of *Arabidopsis AtMAPKs*, we comprehensively analyzed the evolutionary relationship between *LsMAPKs* and *AtMAPKs* (Figure 1). Based on the analysis, we systematically renamed them according to the phylogenetic relationship (*LsMAPKs* in Table 1). Among them, *LsMAPK4* and *LsMAPK4** are identical genes (Appendix A). The corresponding gene name, gene registration number, chromosome position, isomer numbers, protein size, molecular weight, isoelectric point (*pI*), subcellular localization, and transmembrane domain are summarized in Table 1. The lengths of the LsMAPK proteins ranged from 284 (LsMAPK4-3) to 782 (LsMAPK16-4) amino acids, with a molecular weight ranging from 33.00 to 89.70 kDa. The isomer numbers varied from 1 to 4, and the theoretical *pI* value scope was from 4.74 (LsMAPK4-3) to 9.08 (LsMAPK16). The subcellular localization of all MAPKs was located in the nucleus without a transmembrane domain. Chromosome mapping results showed that the 17 *LsMAPKs* were unevenly distributed on 8 lettuce chromosomes, ranging in number from 1 to 3, with none on chromosome 6 (Table 1, Figure 1).

### 2.2. Phylogenetic Relationships and Multiple Sequence Alignment Analysis of LsMAPKs

To analyze the phylogenetic relationship and classification of *LsMAPKs*, a phylogenetic tree was constructed based on *AtMAPKs* of *Arabidopsis thaliana* and *LsMAPKs* of lettuce (Figure 2). *LsMAPK* members were clustered into four groups: A, B, C, and D, which included 3 (*LsMAPK3/3-2/6*), 4 (*LsMAPK4/4*/4-2/4-3*), 4 (*LsMAPK1/1-2/1-3/7*), and 6 (*LsMAPK9/9-2/16/16-2/16-3/16-4*) *LsMAPKs*, respectively. Multisequence alignment analysis showed that all *LsMAPKs* contained a highly conserved tripeptide motif of TXY (Appendix A). Groups A, B, and C contain TEY, and group D contains TDY. Group A and B members also contain the conserved CD domain in (LH)DXXDE(P)XC, which is considered to be a binding site for MAPKK. In addition, more plant species were used to analyze the differentiation of the MAPK gene family in plants (Figure 3), including *Physcomitrella patens*, *Chlamydomonas reinhardtii*, *Selaginella moellendorffii*, *Arabidopsis thaliana*, *Oryza sativa*, *Zea mays*, *Solanum lycopersicum*, and *Populus trichocarpa*. All MAPK genes fell into five groups, and the new group of MAPKs (group O) was mainly composed of lower eukaryotic and fern members, which were *CreinMAPK4-1*, *CreinMAPK7*, and *SmMAPK10*. Compared with higher eukaryotic angiosperms, lower plants encode very few MAPK genes. Among mosses, algae, and ferns, *C. reinhardtii* lacks the ‘group B’ type of MAPKs. All three species lacked the ‘group A’ type. It can also be seen from the phylogenetic tree that group A had the fewest members and group D had the most members.

### 2.3. Gene Structure and Motif Location Analysis of LsMAPKs

We conducted analyses to characterize the sequences of the 17 *LsMAPKs* by motif distribution, conserved domain, and exon–intron structure (Figure 4). Fifteen distinct motifs were predicted (Figure 4B), among which motifs 1-8 were highly conserved and appeared in almost the same order and position in all LsMAPKs proteins, although LsMAPK16 lacked motif 8 and LsMAPK9-2 lacked motifs 5 and 8. Motifs 9 and 10 were unique to groups A and B, and motif 15 was specific to group D. Compared with other group D members, LsMAPK16, LsMAPK16-2, and LsMAPK16-4 specifically had motifs 11-14. Conservative domain analysis showed that all LsMAPKs proteins had the same PKc-like superfamily domain (Figure 4C), in which groups A, B, and C generally had the STKc_TEY_MAPK domain, whereas group D generally had the STKc_TDY_MAPK domain. LsMAPK16, 16-2, and 16-4 had one additional domain. The results of exon–intron structure analysis showed that the number of exons and introns in groups A, B, and C was relatively conservative (Figure 4D). In contrast, the number of introns and exons in group D was distinctly different from that in groups A, B, and C, and the numbers in Group D were generally greater than those in groups A, B, and C.

### 2.4. Synteny Analysis of LsMAPKs

Collinearity analysis can further reveal the potential evolutionary mechanism of species’ gene families. Therefore, the evolutionary relationship and homology between *LsMAPK* members, lettuce and *Arabidopsis*, and lettuce and tomato were predicted. Among the 4291 collinear pairs in lettuce, five collinear pairs belong to *LsMAPKs*, among which *LsMAPK1-2* was collinear with *LsMAPK1-3*, *LsMAPK1*, and *LsMAPK7* at the same time (Figure 5). Furthermore, the comparison of lettuce with *Arabidopsis* and tomato at the genomic level showed that 12,492 collinear pairs were obtained between lettuce and *Arabidopsis*, and 15,558 collinear pairs were obtained between lettuce and tomato (Figure 6), among which the numbers belonging to MAPK members were 8 and 13, respectively. The collinearity pairs of *LsMAPK1*, *4*, and *7* accounted for a total of 12 pairs, indicating the substantial role of these genes in evolution.

### 2.5. Cis-Acting Element Analysis of LsMAPKs

After cis-acting element analysis of the *LsMAPK* promoter region, a total of 64 types of elements were identified and divided into four subgroups: phytohormones, plant development, stress responsiveness, and light responsiveness (Figure 7). The members of each subgroup were evenly distributed on all MAPKs, with the most elements related to stress response and the fewest elements related to development (Figure 7A–D). Among a total of 706 cis-acting elements, MYB elements occurred most frequently at 101 times, followed by MYC elements, which appeared 77 times (Figure 7E). However, MYC elements were absent from *LsMAPK16-4*. Members with close relatives may not show similar types of cis-acting elements. Here, for example, *LsMAPK16* in group D, *LsMAPK3* in group A, and *LsMAPK4* and *4** in group B were classified into the same group according to the type and number of cis-acting elements, suggesting that they may be closely related in function.

### 2.6. Phosphorylation Site Analysis of LsMAPKs

Activation of protein kinases is usually achieved by phosphorylation, so the analysis of phosphorylation sites is vital to understanding the mechanisms of protein action. The predicted phosphorylation sites of *LsMAPK* gene family members showed that a total of 1131 reliable phosphorylation sites (the same site could be phosphorylated by multiple kinases) were widely distributed in the sequence (Figure 8A). The number of nonspecific sites (phos-unsp) was the largest, with a total of 436, followed by phosphokinase C-specific sites (phos-PKC), with a total of 154, and casein kinase II sites, with a total of 105 (Figure 8B). Only one specific phosphorylation site of protein kinase B (phos-PKB) was predicted in *LsMAPK16* and *16-2*. *LsMAPK16*, *16-2*, and *16-4* had the most nonspecific and total phosphate sites, which may be attributed to their sequence length. However, the number of phosphorylation sites was not necessarily related to the length of the amino acid sequence. *LsMAPK4-3* had the shortest amino acid sequence length, but its phosphorylation sites were still more abundant than those of *LsMAPK1*, *3*, and *7*.

### 2.7. Expression Pattern Analysis of LsMAPKs in Lettuce under High Temperature

To investigate the potential regulatory mechanism of the *MAPK* gene in high-temperature bolting, the expression patterns of *LsMAPKs* were quantitatively assessed in response to high temperature in the stems of lettuce (Figure 9). Our previous research demonstrated that the experimental material GB-30 generally began bolting on the 8th day after high-temperature treatment. The results showed that the 16 *LsMAPKs* demonstrated a specific expression pattern during high-temperature bolting in lettuce.

The expression levels of *LsMAPK1*, *LsMAPK1-3*, *LsMAPK4-2*, and *LsMAPK7* on the 8th day were significantly lower than those of the control. Among them, no difference was found on the 16th day in *LsMAPK1*, *LsMAPK1-3*, and *LsMAPK7*. On the 24th day, the expression of *LsMAPK1* and *LsMAPK7* was lower, but that of *LsMAPK1-3* was higher. The expression of *LsMAPK4-2* remained lower than that of the control after HT treatment. The transcriptional levels of *LsMAPK1-2*, *LsMAPK4-3*, and *LsMAPK16* were higher than those of the control only on the 16th day, showing no difference on the 8th and 24th days after HT treatment.

*LsMAPK3-2*, *LsMAPK4*, and *LsMAPK6* were abundantly expressed after HT treatment. *LsMAPK3-2* peaked on the 8th day, and *LsMAPK4* and *LsMAPK6* peaked on the 24th day. The expression level of *LsMAPK9-2* was higher than that of the control on the 8th and 24th days and showed no difference on the 16th day. *LsMAPK16-2*, *LsMAPK16-3*, and *LsMAPK16-4* presented similar expression patterns. Their expression levels were substantially higher than those of the control on the 8th and 16th days, and except that *LsMAPK16-4* was lower, there was no difference on the 24th day. In addition, the expression level of *LsMAPK3* was higher than that of the control on the 8th day, then decreased, and was lower than that on the 16th and 24th day. Overall, qRT–PCR analysis suggested that *LsMAPKs* had a high likelihood of involvement in the bolting of lettuce induced by high temperature.

### 2.8. VIGS-Induced LsMAPK4 Silencing Delayed Bolting of Lettuce under High Temperature

In the previous comparative proteomics, we found that among LsMAPKs proteins, LsMAPK3/4/9 were differentially expressed in high-temperature bolting of leaf lettuce [47]. The subsequent qRT–PCR indicated that the transcription level of *LsMAPK4* among the above three *LsMAPKs* was markedly and continuously upregulated during high-temperature bolting. To further explore gene function, VIGS-mediated *LsMAPK4* gene silencing in lettuce was conducted.

A 355 bp fragment of *LsMAPK4* was successfully cloned into the pTRV2 vector (Figure 10A). Three weeks after Agrobacterium infection, RT–PCR amplification results showed that the pTRV2 empty vector and pTRV2-*LsMAPK4* vector had been transferred into lettuce and effectively expressed (Figure 10B,C). Then, to finally determine whether *LsMAPK4* was silenced, the expression was measured by qRT–PCR (Figure 10D). The results showed that the expression of *LsMAPK4* in the silenced group was significantly lower than that in the blank control group and the empty vector group.

In the third week after the VIGS infection, the three groups of lettuce were treated with HT. There was no significant difference in stem length among the three groups at 0 and 2 days after HT treatment (Figure 10E). On the 4th day, the stem length of the silent group began and continued to be significantly shorter than that of the blank control group and the empty vector group. On the 8th day, the stems of the blank control and the empty vector group elongated rapidly, and both of them began bolting (Figure 10F). At this time, the flower bud differentiation status of the stem tips of the three groups was detected by paraffin sectioning to judge whether the lettuce was bolting (Figure 10G). The results showed that the growth cone of the stem tips of the blank control and the empty vector group was round, blunt, and hypertrophic, without obvious protuberance, and had entered the early stage of flower bud differentiation. However, the growth cone of the silent group was semicircular with obvious protuberances, indicating that it was still in the vegetative growth stage and had not yet entered the early stage of flower bud differentiation to start bolting. Overall, silencing *LsMAPK4* significantly delayed the bolting of lettuce under high temperatures, indicating that *LsMAPK4* may play a promoting role in high-temperature bolting in lettuce.

## 3. Discussion

The MAPK cascade is a functional module that widely exists in eukaryotes [48] and has been well studied in *Arabidopsis*, rice, and other plants. However, there are few reports regarding its occurrence in lettuce. Our study identified a total of 17 MAPK family members in the lettuce genome, of which *LsMAPK4* and *LsMAPK4** were found to be identical. The 17 MAPK family members found in lettuce is fewer than the number in *Arabidopsis* (20 genes) [49], corn (19 genes) [50], and poplar (21 genes) [51] but more than the number in tomato (16 genes) [52], *C. reinhardtii* (6 genes), *P. patens* (10 genes), and *S. moellendorffii* (6 genes) [53] and the same as the number found in rice [51]. This indicates that there is no correlation between the number of MAPKs in plants and a plant species’ genome size. The chromosome distribution pattern of the MAPK family indicates that they are located on chromosomes individually or in clusters, but they do not occur on all chromosomes. None of the *LsMAPK* family members are located on chromosome 6, just as no MAPK members are distributed on chromosome 5 of *Brachypodium distachyon* [54] or chromosome 3 of cucumber [55].

Through phylogenetic analysis and multiple sequence alignment, *LsMAPK* family members can be divided into four groups according to the TXY motif. To study the molecular evolution and phylogenetic relationship of MAPKs in lettuce, we conducted phylogenetic analysis on 132 MAPKs from different plant groups. In lettuce, dicotyledons, and monocotyledons, MAPKs in group A have orthologs and paralogs of MAPK3 and MAPK6 but no orthologs of MAPK10. MAPK3 and MAPK6 are either present together or absent, and the decisive functions of these two MAPKs in growth and development and response to biotic and abiotic stresses have been confirmed in various plants [41,56,57,58], implying that MAPK3 and MAPK6 orthologs are indispensable. Another member of group A, the ortholog of MAPK10, was identified only in mustard plants [53], indicating that the ortholog of MAPK10 is only conserved in the Brassicaceae family of dicotyledons and has been lost from other families during evolution. In group B, MAPK5 and MAPK11 are similar to MAPK10, and MAPK11 has been lost even in the mustard family. In lettuce, all group B *LsMAPKs* are orthologs or paralogs of MAPK4, and there are two identical *LsMAPK4s*, indicating that a wide range of repetitive events may occur in its genome. *LsMAPKs* in group C exist in the form of MAPK1 and MAPK7, similar to the tomato genome. Group D is a type of MAPK unique to plants. None of the studied species lacks group D MAPKs. With the increase in biological complexity, the group D MAPKs expand through gene replication [22], which is indispensable in the green plant lineage. Collinearity analysis further revealed the potential evolutionary mechanism of the species gene family. There were five collinear pairs among *LsMAPK* members of lettuce and eight pairs among *AtMAPKs* of *Arabidopsis thaliana*. There were 13 pairs of *SlMAPKs* in tomato. This indicated that the genetic relationship between tomato and lettuce was closer than that of *Arabidopsis*.

Gene structure is closely related to gene expression and function. All LsMAPKs proteins contain motifs 1-7 (except LsMAPK4-3, which lacked motif 3). The C-terminal motif is more conserved than the N-terminal motif. There are unique characteristic motifs among distinct groups, indicating their evolutionary relationship and different functional divisions [59]. Transcript information indicates that *LsMAPKs* have different exon arrays in their genes. The distribution of introns and exons in the gene structure of *LsMAPKs* showed regularity among diverse groups. Both group A and group B contain 6 exons, group C contains 2–3 exons, and group D contains more exons, including 8–12 introns. Similar structural patterns were also observed in other plants, with high conservation within groups and an elevated level of variation between groups [49,53]. *SmMAPK10* of *S. moellendorfii* contains up to 15 exons, and some intron-free MAPKs also exist in higher eukaryotic plants, such as *PaMAPK2*, *PaMAPK3*, *PaMAPK7-1*, and *PaMAPK20* of *Picea abies* [53]. The plant MAPK cascade pathway mediates the response to salt, drought, cold, high temperature, and pathogenic bacteria [5] and is cross-linked with the hormone signaling pathway to regulate plant growth and development [60,61]. The promoter of the *LsMAPK* gene is rich in various cis-acting elements, such as ABA response elements (ABREs), MYB elements, MYC elements, methyl jasmonate reaction motifs (TGACG motifs and CGTCA motifs), root-specific expression elements (as-1 s), and various light response elements, which is strong evidence that *LsMAPKs* may participate in various life activities. Analysis of phosphorylation sites is very important for understanding the mechanism of action of the LsMAPK protein kinase. In addition to nonspecific sites, phosphokinase C (phos-PKC), casein kinase II (phos-CKII), and cell division cyclin 2 (phos-cdc2) have the largest number of specific phosphorylation sites, which have been reported to participate in plant life activities such as stress resistance, cell apoptosis, flowering regulation, and cell division [62,63,64,65]. It is speculated that *LsMAPK* is related to growth and development and the regulation of stress resistance.

Plants suffer from various adversity stresses during their growth and development. Among them, high-temperature stress affects almost all aspects of plant growth, development, reproduction, and yield, resulting in shedding, flowering, fruit abortion, and even whole plant death [15,66,67]. Activation of MAPK plays a key role in the perception and transduction of temperature signals [68]. MAPKs in *B. distachyon* are temperature-sensitive, 60% of which are induced by high temperature [69]. High temperature can induce the activity of *AtMAPK6* in *Arabidopsis*, but it has also been reported that *AtMAPK6* negatively regulates the high-temperature response [70,71]. Most cucumber *CsMAPKs* (except *CsMAPK3* and *CsMAPK7*) were upregulated after high-temperature treatment [55]. In chrysanthemum, except for the increased expression levels of *CmMAPK4.1*, *CmMAPK6*, and *CmMAPK13* after 1 h of heat shock treatment, the other MAPKs decreased or remained unchanged [24]. The high-temperature treatment times used in the above literature were mostly between 1 and 8 h. In this paper, the changes *in LsMAPK* expression in lettuce after high-temperature treatment were continuously monitored and detected at 0, 8, 16, and 24 days after high-temperature treatment with a higher sampling frequency and longer observation time. Among them, the most obvious were *LsMAPK3-2*, *LsMAPK4*, and *LsMAPK6*, which were significantly and continuously higher than the control after high temperature. *LsMAPK4-2* was significantly and continuously lower than the control, and the rest of the *LsMAPKs* had a tendency to decrease or to rise first and then decrease compared with the control, but not every stage had a significant difference.

At present, studies on MAPK4 gene function mainly focus on cell division and stress response. Tomato MAPK4 [72], rice MAPK4 [73], and *Arabidopsis* MAPK4 [74] have been reported to play a role in low-temperature stress. In addition, MAPK4 was more reported to be related to cell division, flower organ formation, and photosynthesis [18,37,75], and little has been reported about the high-temperature stress response. In our study, we screened *LsMAPK*4 and *LsMAPK*4* on chromosome 3, which are located in different positions but are very close to each other, with a distance of about 180,000 bp. Their gene sequences and open reading frame sequences are identical. No same gene was found in the front and back of the two chromosome positions, indicating that this is a small and segmental duplication event. Compositae species have experienced the triplication of the whole genome in the evolutionary process, and leaf lettuce also shows the triplication of the genome, which leads to a large number of replication events [76]. The segmental duplication of *LsMAPK4* is probably the result of the retention of duplicate copies. We have noticed that the promoter sequences of *LsMAPK4* and *LsMAP**K4** were completely different, and there were differences in cis-acting elements (Figure 7), which may lead to functional differences between the two genes. But it is also very likely that the functions of the two genes will overlap under a specific condition. The gene replication of *LsMAPK4* is likely to make its expression more extensive and efficient. Since the expression of *LsMAPK4* was significantly upregulated by high temperature, we further conducted VIGS experiments to explore the function of *LsMAPK4* in the high-temperature bolting of lettuce. According to the cytological and morphological experiments of flower buds and stems, after 8 days of high-temperature treatment, bolting occurred in the blank control group and the empty vector group but not in the silenced group. Easy-bolting lettuce GB-30 was used as the experimental material. After high-temperature treatment, GB-30 began bolting on the 8th day [46]. This is very consistent with the VIGS phenotype. High temperature significantly promoted the expression of *LsMAPK4*, which played a positive regulatory role in the bolting process of lettuce at high temperatures.

## 4. Materials and Methods

### 4.1. Genome-Wide Identification of LsMAPK Family Members in Lettuce

Whole-genome data for lettuce (https://phytozome.jgi.doe.gov/pz/portal.html; accessed on 6 August 2021) were downloaded and constructed into a local protein database. The *Arabidopsis thaliana* MAPK protein sequences from the *Arabidopsis thaliana* genome database (http://www.arabidopsis.org/; accessed on 6 August 2021) were used as the search seed for a BLASTP homology search of lettuce whole-genome data with an e-value of 1 × 10^−5^ and a minimum identity of 50% as a threshold. The HMMER3.0 program (http://hmmer.org/; accessed on 10 September 2021) was used to remove redundant sequences by applying the serine/threonine protein kinase-like domain (PF00069) as a query for hidden Markov model (HMM) searches with a threshold of 1 × 10^−5^.

### 4.2. Bioinformatics Analysis of LsMAPKs

By querying the annotation file of the lettuce genome, the chromosome location, direction, and isomer number of MAPK proteins were obtained, and the chromosome location was visualized by TBtools. The amino acid number, isoelectric point, and molecular weight of the MAPK protein were predicted by using the ProParam tool (https://web.expasy.org/protparam/; accessed on 5 July 2022). Cell-PLoc 2.0 was used to predict subcellular localization (http://www.csbio.sjtu.edu.cn/bioinf/Cell-PLoc-2/; accessed on 5 July 2022). TMHMM was used to predict transmembrane domains (https://dtu.biolib.com/DeepTMHMM; accessed on 5 July 2022). The MAPK members of each species were obtained from the corresponding plant database, and the species database was downloaded from Phytozome V13 (https://phytozome-next.jgi.doe.gov/; accessed on 11 July 2022). The construction of the phylogenetic tree was completed by MEGA 7.0 software and enhanced using Evolview (http://www.evolgenius.info/evolview/#/login; accessed on 13 July 2022). The conserved domains of proteins were analyzed by NCBI-CD search (https://www.ncbi.nlm.nih.gov/Structure/cdd/wrpsb.cgi; accessed on 12 July 2022). MEME Suite analyzed conserved motifs of proteins (http://meme-suite.org/tools/meme; accessed on 12 July 2022). The cis-acting elements were analyzed by Plant CARE (http://bioinformatics.psb.ugent.be/webtools/plantcare/html/; accessed on 6 July 2022). Analysis of phosphorylation sites was completed by using netphos (https://services.healthtech.dtu.dk/service.php?NetPhos-3.1; accessed on 24 July 2022). Intron–exon structure, whole genome collinearity analysis of lettuce, and collinearity analysis of lettuce with *Arabidopsis* and tomato were analyzed by TBtools. The visualization of all the above analyses was completed by TBtools and further enhanced using Adobe Illustrator.

### 4.3. Plant Materials and Treatments

Easily bolting lettuce variety ‘GB30’ was used as the experimental material, which was from seeding plant protection Co., LTD in Inner Mongolia Bameng Fidelity, conserved in our laboratory. It was planted in a matrix of sand, soil, and peat (1:1:1 by volume) and cultured in the computer greenhouse of Beijing Agricultural College under the following conditions: temperature of 20 ± 2 °C (day)/13 ± 2 °C (night), photoperiod of 14 h (day)/10 h (night), relative humidity of 60 ± 5%, and light intensity of 12,000 lux. The plants were transplanted at the trefoil stage, and high-temperature treatment was carried out when lettuce had six true leaves [46]. The control group maintained the original culture conditions, whereas the high-temperature group increased the temperature to 33 ± 2 °C (day)/25 ± 2 °C (night), with the other conditions unchanged. Samples were taken at 0, 8, 16, and 24 days of treatment, and the sampling site was the stem. Every five plants were used as a replicate, and the sampling was repeated three times independently.

### 4.4. RNA Extraction and Quantitative Real-Time PCR Analysis (qRT–PCR)

Total RNA was extracted using a Quick RNA Isolation kit (Huayueyang Biotech, Beijing, China). The first-stand cDNA of lettuce was synthesized by *TransScript*^®^ Uni All-in-One First-Strand cDNA Synthesis SuperMix (TransGen Biotech, Beijing, China). Primer3 Plus software was used to design qRT–PCR primers (Appendix A), and 18S rRNA (HM047292.1) was used as an internal reference gene. qRT–PCR was performed using TB Green^®^ Premix Ex Taq™ II (Takara Bio, Beijing, China) and a CFX96 Touch Real-Time PCR Detection System (Bio-Rad Laboratories, Hercules, CA, USA). The amplification program was as follows: pre-denaturation at 95 °C for 3 min, denaturation at 95 °C for 20 s, and annealing at 55 °C for 30 s for 40 cycles. The 2^−ΔΔCT^ method was used to analyze the data.

### 4.5. Virus-Induced Gene Silencing Analysis

A 355 bp sequence in the CDS region of *LsMAPK4* was amplified by VIGS-*LsMAPK4*-F/R primers (Appendix A). The pTRV2 vector was linearized by EcoRI and XhoI. A ClonExpress Ⅱ One Step Cloning Kit (Vazyme Biotech, Beijing, China) was used to complicate the homologous recombination of the amplified band and linearized vector. After the recombinant plasmid was transferred into *Trans*1-T1 Phage Resistant Chemically Competent Cells and sequenced correctly, it was transferred into GV3101 Chemically Competent Cells (Weidi Biotechnology, Shanghai, China).

Before VIGS injection, the same amount of pTRV11 and pTRV2 bacterial solution was mixed. There were 30 lettuces in each group, and there were three treatment groups in total: the blank control group (no injection), empty vector group (injection with equal pTRV2 and pTRV1), and silent group (injection with equal pTRV2-*LsMAPK4* and pTRV1). Three weeks after injection, a high-temperature treatment was carried out. Before treatment, RNA from new leaves was taken to detect whether the vector was transferred and expressed, and the expression level of *LsMAPK4* was measured. The vector detection primers were pTRV2-F/R for the empty vector group and VIGS-*LsMAPK4*-F/R for the silencing group (Appendix A). The stem length of each group was measured at 0, 2, 4, 6, 8, and 10 days after high temperature, and the stem tips of the plants were paraffin-sectioned on the 8th day.

### 4.6. Statistical Analysis

All tests were performed in triplicate. One-way ANOVA was performed on the data using statistical analysis software SPSS 12.5 (International Business Machine, Chicago, IL, USA), and graphs were drawn by Origin 9 (Origin Lab, Northampton, MA, USA). The standard error is indicated. The * stands for *p* < 0.05 and the ** stands for *p* < 0.01, followed by Student’s *t*-test. Different letters represent significant differences as determined using one-way ANOVA followed by Duncan’s test. *p* < 0.05.

## 5. Conclusions

In this study, we identified a total of 17 MAPK gene family members from the whole genome of lettuce. The physicochemical properties, chromosomal localization, phylogeny, gene structure, family evolution, cis-acting elements, and phosphorylation sites were comprehensively analyzed. To explore the role of *LsMAPKs* in high-temperature bolting of lettuce, the expression pattern of *LsMAPKs* was evaluated within 24 days after high-temperature treatment, and the positive regulatory role of *LsMAPK4* in high-temperature bolting was verified by VIGS. This provides a theoretical basis for understanding high-temperature bolting mechanisms in lettuce and identifies potential paths for improving the annual production of lettuce.

## Figures and Tables

**Figure 1 ijms-23-11129-f001:**
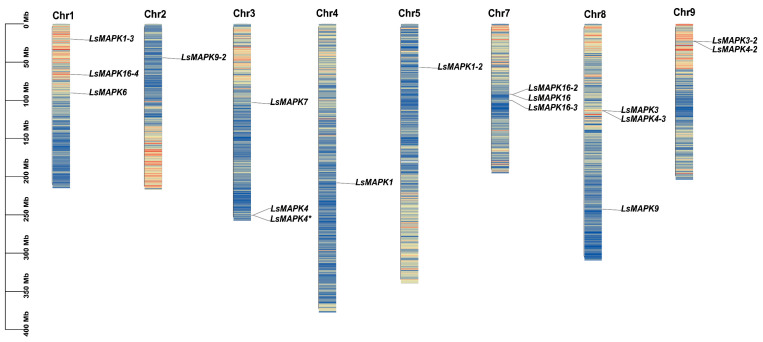
Chromosomal location of *LsMAPKs*. The chromosome number is indicated at the top, and the scale on the left represents the chromosome size. Without *LsMAPK* localization, chromosome 6 is not shown.

**Figure 2 ijms-23-11129-f002:**
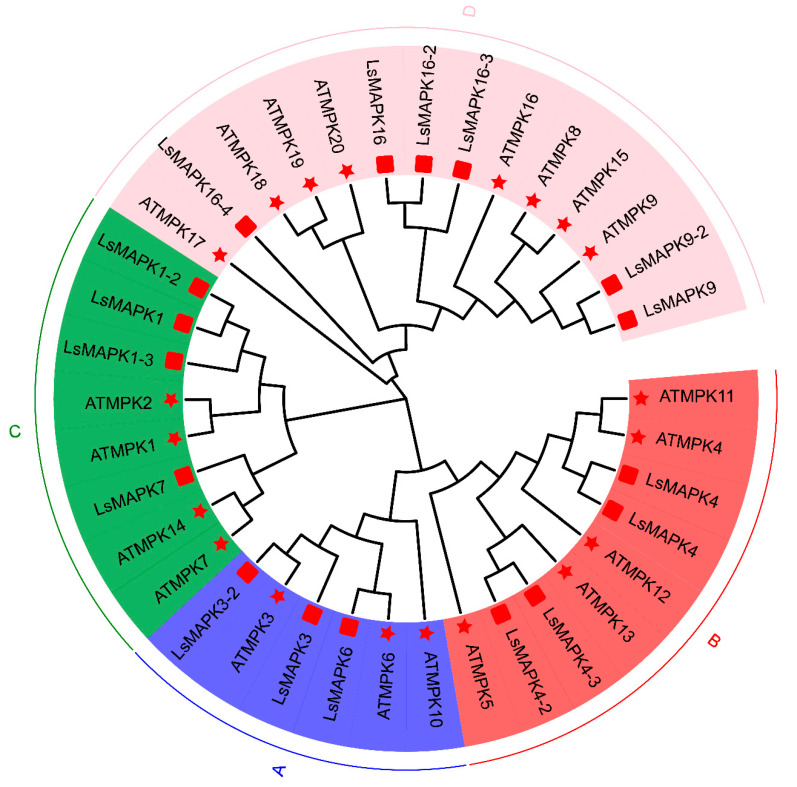
Phylogenetic tree analysis of the MAPK gene family from lettuce and *Aradiopsis*. Distinct color blocks represent different groups, five-pointed stars represent lettuce, and squares represent *Arabidopsis*.

**Figure 3 ijms-23-11129-f003:**
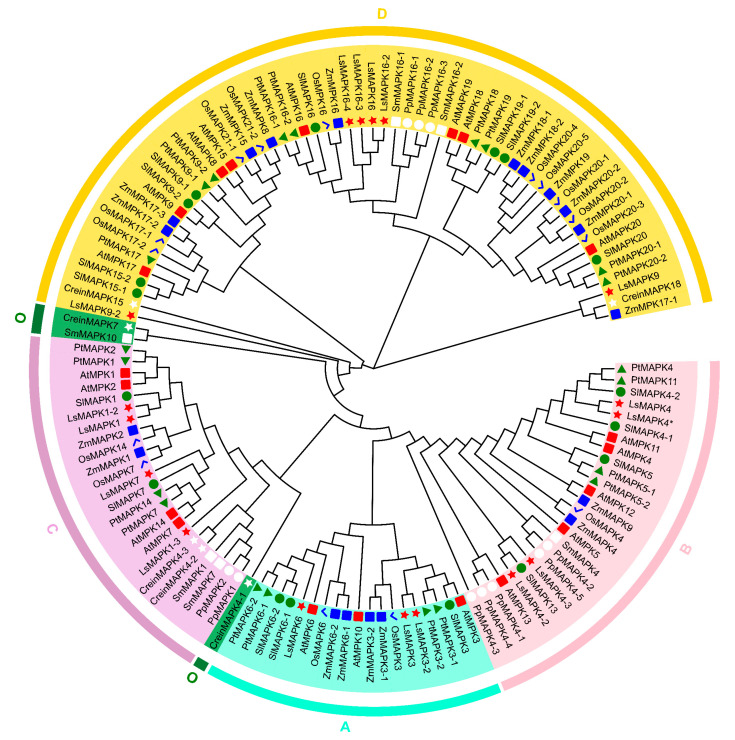
Phylogenetic relationships of the MAPK gene family from *Physcomitrella patens* (*PpMAPKs*), *Chlamydomonas reinhardtii* (*CreinMAPKs*), *Selaginella moellendorffii* (*SmMAPKs*), *Arabidopsis thaliana* (*AtMAPKs*), *Oryza sativa* (*OsMAPKs*), *Zea mays* (*ZmMAPKs*), *Solanum lycopersicum* (*SlMAPKs*), *Populus trichocarpa* (*PtMAPKs*), and *Lactuca sativa* (*LsMAPKs*). Distinct color blocks represent different groups. The different shapes and colors of the symbols indicate different species.

**Figure 4 ijms-23-11129-f004:**
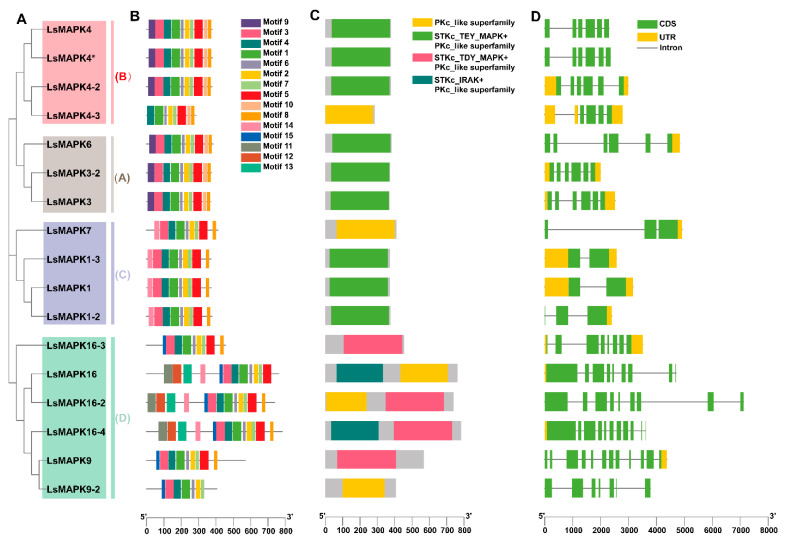
Gene structure analysis of *LsMAPKs*. *LsMAPK* gene family phylogenetic tree (**A**), motif distribution (**B**), conservative domain analysis (**C**), and intron–exon structure (**D**).

**Figure 5 ijms-23-11129-f005:**
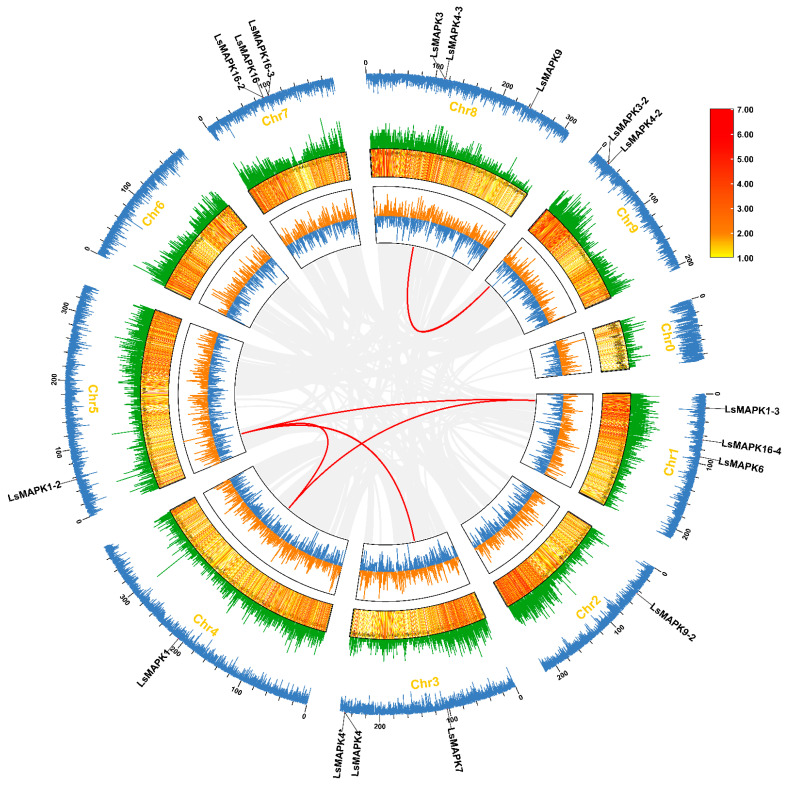
Collinearity analysis of the *LsMAPK* gene family. The gray line in the background represents the homolinear pair in the whole lettuce genome, and the red line represents the homolinear pair of *LsMAPKs*. The information represented by each circle in the figure is GC skew, genome density heatmap, GC radio, genome density linear map, chromosome name, N radio, chromosome length scale, and *LsMAPKs* chromosome location annotation from inside to outside.

**Figure 6 ijms-23-11129-f006:**
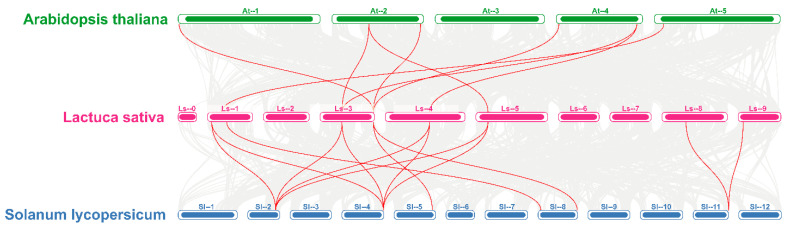
Collinearity analysis of lettuce with *Arabidopsis* and tomato. The gray line in the background represents all collinear pairs of lettuce with *Arabidopsis* and tomato at the genomic level. The red line represents the collinear pairs belonging to MAPK family members from those three species.

**Figure 7 ijms-23-11129-f007:**
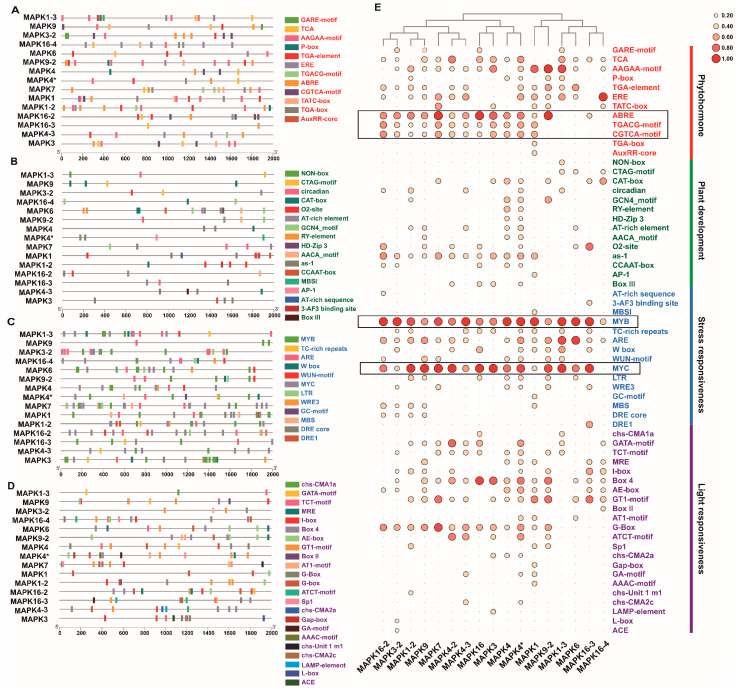
Promoter analysis of *LsMAPKs*. Location diagram of cis-acting elements related to phytohormones (**A**), plant development (**B**), stress responsiveness (**C**), and light responsiveness (**D**). (**E**) Quantity heatmap of all instantaneously acting elements.

**Figure 8 ijms-23-11129-f008:**
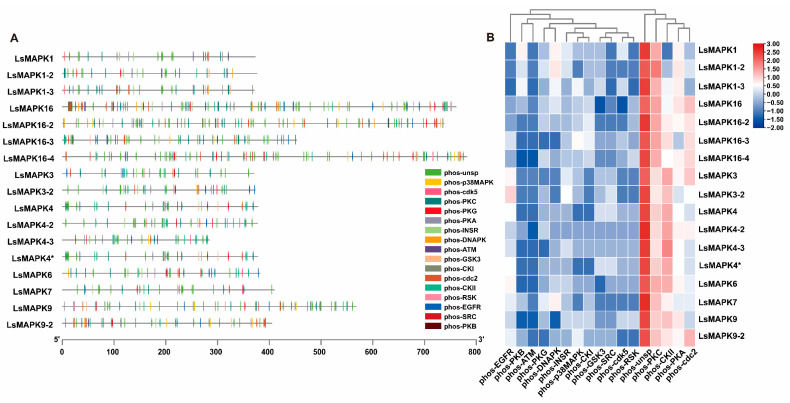
Prediction of phosphorylation sites of *LsMAPKs*. (**A**) Distribution of phosphorylation sites of protein kinases in *LsMAPKs*. (**B**) Quantitative heatmap of protein kinases at phosphorylatable sites.

**Figure 9 ijms-23-11129-f009:**
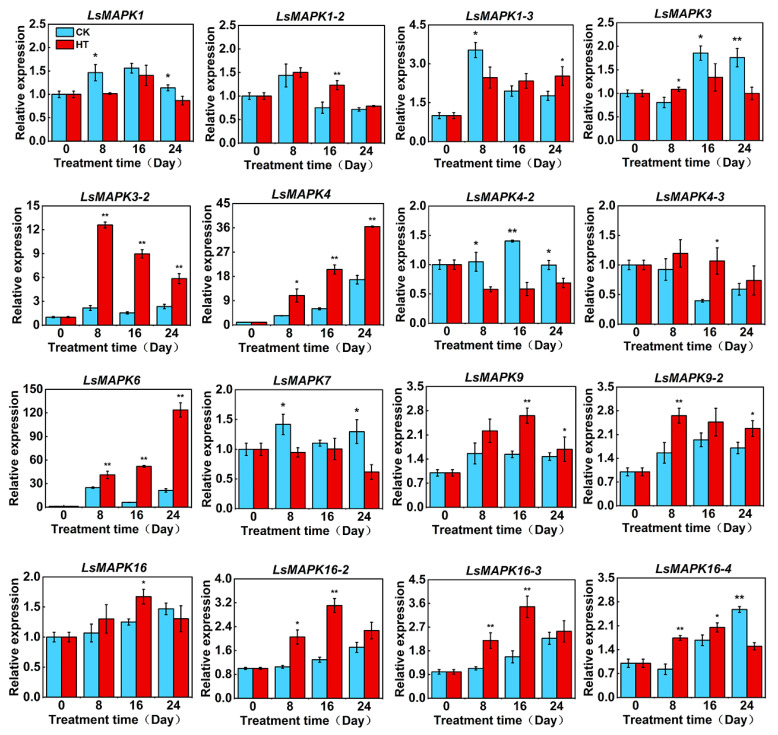
Expression analysis of *LsMAPKs* in lettuce under high temperature. HT, high temperature. The standard error is indicated, * stands for *p* < 0.05, and ** stands for *p* < 0.01.

**Figure 10 ijms-23-11129-f010:**
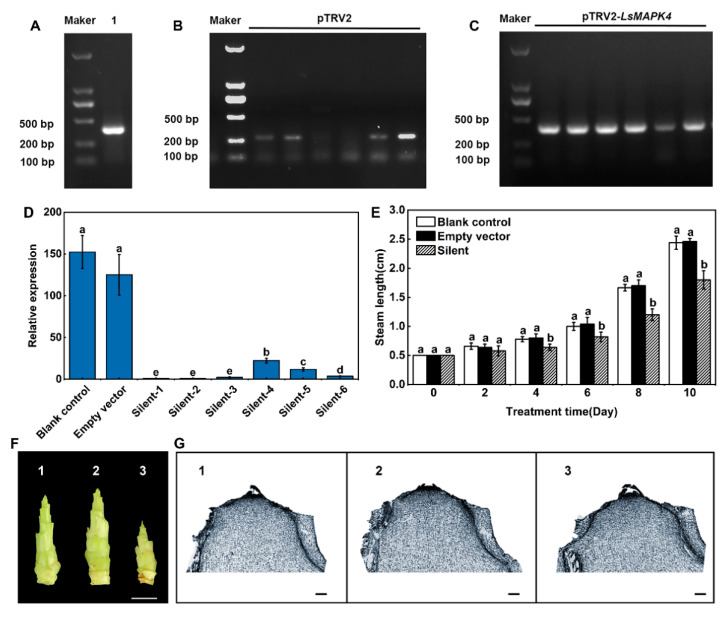
Effect of VIGS-mediated *LsMAPK4* gene silencing on high-temperature bolting of lettuce. (**A**) Construction of the pTRV2-*LsMAPK4* vector. (**B**) Detection of pTRV2 fragments. (**C**) Detection of pTRV2-*LsMAPK4* fragments. (**D**) Expression of *LsMAPK4* in the blank control group, the empty vector group, and the silenced group. (**E**) Changes in stem length of VIGS-infected plants after HT treatment. (**F**) Morphology of the stem on the 8th day after HT. Bar = 0.5 cm. (**G**) Morphology of the shoot tip on the 8th day of HT. Bar = 200 µm. 1. the blank control group; 2. the empty vector group; 3. the silent group. The standard error is indicated. Different letters represent significant differences as determined using one-way ANOVA followed by Duncan’s test. *p* < 0.05.

**Table 1 ijms-23-11129-t001:** Physical and chemical aspects of *LsMAPKs*.

Gene Name	Locus ID	Chromosomal Location				Gene Models	Putative Proteins			Subcellular Localization	Transmembrane Domain
		Chr	Chr_start	Chr_end	Direction		Length(aa)	MW (kDa)	pI		
** *LsMAPK1* **	**Lsat_1_v5_gn_4_117181**	**4**	**208105012**	**208108237**	**−**	**2**	**373**	**43.08**	**6.50**	**Nucleus**	**none**
** *LsMAPK1-2* **	**Lsat_1_v5_gn_5_26820**	**5**	**56806062**	**56808833**	**−**	**2**	**376**	**43.48**	**7.62**	**Nucleus**	**none**
** *LsMAPK1-3* **	**Lsat_1_v5_gn_1_17080**	**1**	**20023975**	**20026544**	**−**	**1**	**371**	**42.42**	**5.83**	**Nucleus**	**none**
** *LsMAPK3* **	**Lsat_1_v5_gn_8_76860**	**8**	**113432000**	**113434556**	**−**	**3**	**370**	**42.60**	**5.62**	**Nucleus**	**none**
** *LsMAPK3-2* **	**Lsat_1_v5_gn_9_20480**	**9**	**22563636**	**22565881**	**−**	**2**	**372**	**42.88**	**5.38**	**Nucleus**	**none**
** *LsMAPK4* **	**Lsat_1_v5_gn_3_137680**	**3**	**250231945**	**250234610**	**+**	**2**	**378**	**43.46**	**6.32**	**Nucleus**	**none**
** *LsMAPK4** **	**Lsat_1_v5_gn_3_138401**	**3**	**250423836**	**250426191**	**+**	**1**	**378**	**43.46**	**6.32**	**Nucleus**	**none**
** *LsMAPK4-2* **	**Lsat_1_v5_gn_9_21161**	**9**	**23155272**	**23158265**	**+**	**2**	**377**	**43.46**	**5.86**	**Nucleus**	**none**
** *LsMAPK4-3* **	**Lsat_1_v5_gn_8_76340**	**8**	**114078543**	**114081325**	**−**	**2**	**284**	**33.00**	**4.74**	**Nucleus**	**none**
** *LsMAPK6* **	**Lsat_1_v5_gn_1_74260**	**1**	**90387406**	**90392319**	**+**	**2**	**382**	**43.94**	**5.65**	**Nucleus**	**none**
** *LsMAPK7* **	**Lsat_1_v5_gn_3_77061**	**3**	**103013530**	**103018532**	**−**	**2**	**410**	**47.51**	**5.89**	**Nucleus**	**none**
** *LsMAPK9* **	**Lsat_1_v5_gn_8_144901**	**8**	**242553674**	**242558284**	**−**	**2**	**568**	**64.46**	**8.10**	**Nucleus**	**none**
** *LsMAPK9-2* **	**Lsat_1_v5_gn_2_19180**	**2**	**44466873**	**44470986**	**−**	**3**	**405**	**46.52**	**8.87**	**Nucleus**	**none**
** *LsMAPK16* **	**Lsat_1_v5_gn_7_64441**	**7**	**92076041**	**92080742**	**+**	**1**	**761**	**85.36**	**9.08**	**Nucleus**	**none**
** *LsMAPK16-2* **	**Lsat_1_v5_gn_7_64461**	**7**	**92052037**	**92059155**	**+**	**1**	**738**	**84.03**	**7.15**	**Nucleus**	**none**
** *LsMAPK16-3* **	**Lsat_1_v5_gn_7_66400**	**7**	**100166151**	**100169658**	**−**	**1**	**453**	**51.79**	**6.01**	**Nucleus**	**none**
** *LsMAPK16-4* **	**Lsat_1_v5_gn_1_56481**	**1**	**65962932**	**65970369**	**−**	**5**	**782**	**89.70**	**8.75**	**Nucleus**	**none**

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
