# Peer review of "Genome-Wide Identification and Expression Analysis of MAPK Gene Family in Lettuce (*Lactuca sativa* L.) and Functional Analysis of *LsMAPK4* in High- Temperature-Induced Bolting"

_ijms, 2022, doi:10.3390/ijms231911129_

Round 1
Reviewer 1 Report
This study reported a theoretical basis for further study of specific mechanisms involving the MAPK gene in high-temperature bolting of lettuce.
This study identified LsMAPK family members in the lettuce genome, and most LsMAPKs responded to high temperatures by analysis of the expression patterns. And authors described that silencing of LsMAPK4 could significantly inhibit high temperature-accelerated bolting of lettuce by Virus-induced gene silencing (VIGS), indicating its potential use as a regulator of bolting under thermal stresses.
This report is very interesting.
I recommend accept.
Author Response
“This study reported a theoretical basis for further study of specific mechanisms involving the MAPK gene in high-temperature bolting of lettuce.
This study identified LsMAPK family members in the lettuce genome, and most LsMAPKs responded to high temperatures by analysis of the expression patterns. And authors described that silencing of LsMAPK4 could significantly inhibit high temperature-accelerated bolting of lettuce by Virus-induced gene silencing (VIGS), indicating its potential use as a regulator of bolting under thermal stresses.
This report is very interesting.
I recommend accept.”
Thank you very much for your affirmation of our research work and achievements, and we will make unremitting efforts to improve.
Reviewer 2 Report
The manuscript describing the genome-wide identification of MAPK family in Lettuce is interesting. However, there are number of issues to be addressed before its possible consideration for publication.
1. The authors have not provided any table header or figure legends in the manuscript. This makes the review process difficult and unclear. I urge the authors to provide clear, detailed, and sufficient figure legend for each figure presented in the manuscript.
2. In abstract (line 22-26), the authors are describing about the cis-elements found on the MAPKs. But, this is only a predicted result and also not that significant to be described in abstract. Especially, when the authors can focus on the functional validation of LsMAPK4 and VIGS. I suggest the authors to delete it and add more important lines related to VIGS results. Minor: In line 20, in silicon should be in silico.
3. In line 100, the authors say that LsMPK4 and LsMPK4* are identical. They can include a sequence alignment picture (even as a supplementary file) to support this. Moreover, how the authors have named the MAPKs? Which nomenclature system they have adopted here?
4. The authors have selected LsMPK4 as the candidate gene for VIGS and reasoned that it has shown significant increased expression at all time points. However, other genes have also shown such phenomenon (MPK3-2, MPK6). In fact, MPK6 have shown many fold higher induced expressions as compared to MPK4. So, why the authors did not chose MPK6?
5. Line 240: The control group has been named as blank control, but in the figure it has been shown as black control. Correct it.
6. Although the discussion seems okay, a key observation should have been discussed. The LsMPK4 and LsMPK4* are identical, yet present at two different but very close locations on the same chromosome. What could be the reason of this kind of gene duplication and what could be the effects of it? This should be included in the discussion.
7. In methods, the authors have mentioned ANOVA analysis. They should specify which ANOVA (one-way or two-way).
I reserve my other comments on the manuscript as the figure legends are not presented. Once I get the revised manuscript with proper figure legend, I will include further comments, if necessary.
Author Response
The manuscript describing the genome-wide identification of MAPK family in Lettuce is interesting. However, there are number of issues to be addressed before its possible consideration for publication.
- The authors have not provided any table header or figure legends in the manuscript. This makes the review process difficult and unclear. I urge the authors to provide clear, detailed, and sufficient figure legend for each figure presented in the manuscript.
Thanks for your constructive comments! We have added table and picture legends to the article this time. Actually, we have had prepared the table and picture legends earlier. It may be that there was an error in the first upload, and the table and picture legends were not uploaded successfully. We are very sorry that your review has become difficult and unclear due to our mistakes.
- In abstract (line 22-26), the authors are describing about the cis-elements found on the MAPKs. But, this is only a predicted result and also not that significant to be described in abstract. Especially, when the authors can focus on the functional validation of LsMAPK4 and VIGS. I suggest the authors to delete it and add more important lines related to VIGS results. Minor: In line 20, in silicon should be in silico.
We agree Reviewer 2’s comments. We fully agree that more attention should be paid to the functional verification of LsMAPK4 and VIGS. Therefore, we have added more important lines related to VIGS results. At the same time, we combined the analysis results of cis acting elements and protein phosphate sites and added the collinearity analysis results, which were also important contents in our bioinformatics analysis. In addition, we have corrected the spelling of silicon.
- In line 100, the authors say that LsMPK4 and LsMPK4* are identical. They can include a sequence alignment picture (even as a supplementary file) to support this. Moreover, how the authors have named the MAPKs? Which nomenclature system they have adopted here?
Thanks for your constructive comments! The comparison picture of gene sequence and open reading frame sequence of LsMAPK4 and LsMAPK4* has been taken as supplement Figure S1. Our naming method is based on the unified naming method of Arabidopsis AtMAPK. According to the genetic relationship, we have named the leaf lettuce LsMAPKs. In the revised manuscript, the corresponding part was rewritten to describe the naming method (Please see Page 3, Line102-105).
- The authors have selected LsMPK4 as the candidate gene for VIGS and reasoned that it has shown significant increased expression at all time points. However, other genes have also shown such phenomenon (MPK3-2, MPK6). In fact, MPK6 have shown many fold higher induced expressions as compared to MPK4. So, why the authors did not chose MPK6?
We totally agree reviewer 2’s comments. In the previous experiment, we screened three LsMAPKs with comparative proteomics in high-temperature bolting, namely LsMPAK3, LsMAPK4 and LsMAPK9. In the subsequent qRT-PCR verification, we found that the expression of LsMAPK3-2, LsMAPk6 and LsMAPK4 was continuously up-regulated under high temperature. Comprehensive consideration makes our focus more on LsMAPK4. Actually, LsMAPK6 is also our candidate gene for subsequent functional verification. We have also conducted VIGS experiments and are conducting stable genetic transformation. If the function is significant, we will also publish corresponding articles. In order to clarify the concerns of reviewer 2, we cited the paper on comparative proteomics of high-temperature bolting of leaf lettuce, and modified the corresponding paragraphs accordingly (Please see Page 2, Line89-92 and Page 13, Line258-260).
- Line 240: The control group has been named as blank control, but in the figure it has been shown as black control. Correct it.
Thank you for your careful correction! “Black” in the Figure 10 has been corrected to “blank”.
- Although the discussion seems okay, a key observation should have been discussed. The LsMPK4 and LsMPK4* are identical, yet present at two different but very close locations on the same chromosome. What could be the reason of this kind of gene duplication and what could be the effects of it? This should be included in the discussion.
We compared the 5 'end and 3' end genes of the two genes and found that the genes before and after the two genes were different. At the same time, the distance between them is about 180,000 bp. It indicates that this is a a small segment of segmental duplication event due to the genome triplication of lettuce. The gene replication of LsMAPK4 is likely to make its expression more extensive and efficient, being more functional. Moreover, the corresponding paragraph was rewritten accordingly (Please see Page 16, Line385-398).
- In methods, the authors have mentioned ANOVA analysis. They should specify which ANOVA (one-way or two-way).
One-way ANOVA was performed on the data. The corresponding description was added to the methods 4.6. Statistical Analysis (Please see Page 18, Line 487-489).
I reserve my other comments on the manuscript as the figure legends are not presented. Once I get the revised manuscript with proper figure legend, I will include further comments, if necessary.
Round 2
Reviewer 2 Report
The authors have addressed all my comments in the revised version of the manuscript. However, there are some small mistakes, mostly typos to be fixed before the acceptance of the manuscript.
1. At numerous places (line 24, 183, 184, 188, 213, 406, 409 and so on) and in almost all figure legends, the gene names have to be italicized.
2. In the legend of figure 10, the authors have again mentioned "black control" in stead of "blank control". Please correct this.
3. Please go through the entire manuscript to correct such typos and grammars. A proofread by a non-author is highly advisable.
Author Response
The authors have addressed all my comments in the revised version of the manuscript. However, there are some small mistakes, mostly typos to be fixed before the acceptance of the manuscript.
- At numerous places (line 24, 183, 184, 188, 213, 406, 409 and so on) and in almost all figure legends, the gene names have to be italicized.
Thank you very much for your careful correction. We have italicized the gene names of several places you mentioned. At the same time, we also carefully checked the full text including the legend. When the gene name is used to represent the gene, the gene name is set to italics. When the gene name represents the protein translated by the gene, it is not set as italic.
- In the legend of figure 10, the authors have again mentioned "black control" in stead of "blank control". Please correct this.
Thank you very much for your careful correction. This is a mistake that we have not carefully checked, and it has been corrected.
- Please go through the entire manuscript to correct such typos and grammars. A proofread by a non-author is highly advisable.
Thank you for your constructive suggestions. We have checked and polished our articles with a doctor of relevant major to make our articles more rigorous and meticulous.